# Algae and Cyanobacteria Diversity and Bioindication of Long-Term Changes in the Hula Nature Reserve, Israel

**Sophia Barinova** [1,*] and **Alla Alster** [2]

1    Institute of Evolution, University of Haifa, 199 Abba Khoushi Ave., Mount Carmel, Haifa 3498838, Israel
2    Israel Oceanographic and Limnological Research, The Yigal Allon Kinneret Limnological Laboratory, Migdal 14950, Israel; alster@ocean.org.il
*    Correspondence: sophia@evo.haifa.ac.il; Tel.: +972-4824-97-99

**Abstract:** Lake Hula, the core of one of the most extensive wetland complexes in the Eastern Mediterranean, was drained in 1951–1958. However, about 350 hectares of papyrus marshes were allocated in the southwestern part of the previous lake and became the Hula Nature Reserve status, the first of two wetlands in Israel included in the Ramsar List of Wetlands of International Importance. The list of algae and cyanobacteria species of Lake Hula was compiled by us for the first time based on data from publications of 1938–1958, as well as our research in the Hula Nature Reserve, obtained within the framework of the monitoring program for 2007–2013. The list includes 225 species and intraspecies of algae and cyanobacteria belonging to eight phyla. The dynamics of the species richness of algae and cyanobacteria flora for 1938–2013 are shown. Species-bioindicators of water quality have been identified, and the change in their composition by ecological groups for a period of about a hundred years has been shown. Based on the species richness of algae communities, water quality indices were calculated with particular attention to changes in trophic status during the study period. The algae flora of Lake Hula and Hula Nature Reserve was found to be similar, but bioindication has revealed an increase in salinity and organic pollution in recent years.

**Keywords:** phytoplankton; algae and cyanobacteria flora; indicator species; long-term changes; wetlands; trophic state; Ramsar object; Israel





## 1. Introduction

Wetlands cover at least 6% of the Earth's surface [1]. They play a key role in hydrological and biogeochemical cycles, harbor a large part of the world's biodiversity, and provide multiple services to humankind. However, pressure in the form of land reclamation, intense resource exploitation, changes in hydrology, and pollution threaten wetlands on all continents. Depending on the region, 30–90% of the world's wetlands have already been destroyed or strongly modified in many countries with no sign of abatement [1].

Until the fifties of the last century, Lake Hula was the kernel of the unique Hula wetland ecosystem, one of the most extensive wetland complexes in the Eastern Mediterranean. The wetland complex included the Hula Lake (around 15 km$^2$), extensive peat-based *Cyperus papyrus* swamps, and seasonally inundated inorganic soils in the north part of the lake jointly covering an area of 45–75 km$^2$. The lake and its adjacent swamps provided the habitat with a rich diversity of plants and animals and was a key feeding station for migratory birds [2,3]. Hula Lake and swamps were drained between 1951 and 1958 in an attempt to reduce evapotranspiration, increase the water potential for the whole country, convert 60 km$^2$ of swamps into arable land and use the peat for fertilizer and for the industry. The Jewish National Fund carried out the draining project, yet its objectives were never achieved. Contrary, the draining led to uncontrollable underground fires that resulted in the formation of dangerous caverns within the peat layer and dust storms in the valley [4]. The "World biomes map" (feow.org, accessed on 21 September 2021) [5] reported that Hula Lake was completely drained for region 438. However, a small (350 hectares)

area of papyrus swampland in the southwest of the valley was set aside and, in 1963, became Israel's first nature reserve [2,6]. Hula Nature Reserve was gradually isolated from the center of the historical wetland, and for 30 years, the water quantity and quality provided to the nature reserve totally depended on external factors (such as fishpond needs or the maintenance of the Einan Reservoir, from which water flowed to the nature reserve). In 1994, 100 ha of the least agriculturally productive peat soils in the Hula Valley were re-flooded to create the artificial "Agamon" Lake, located where the most extensive peat deterioration and subsequent winter inundation had occurred. Moreover, a network of 90 km of shallow flood and drainage canals was created to raise the water table level. The old course of the Jordan River was reconstructed as the water supply canal to the lake. At the beginning of the 2000s, due to the poor water quality in the nature reserve and its deteriorating ecosystem, two important changes occurred: significant progress was made toward the recognition of nature's rights for water, and the western channel, which supplies water to the nature reserve, was converted to convey clean Jordan water. The nature reserve's second restoration plan began in 2003–2004, with goals: (1) to create the infrastructure that would provide appropriate water quantities and that would control the water system, and (2) reduce water penetration through the dikes and set a system for monitoring and controlling water [7]. From 1996, Hula Nature Reserve included Ramsar's "List of wetlands of international importance" together with the En Afeq Nature Reserve that only two protected watered areas were established as Ramsar objects in the Eastern Mediterranean [8].

The system of state monitoring of water quality in Israel includes the determination of only some chemical indicators but does not include the determination of the biodiversity of organisms. However, leading Israeli experts in this field carried out biodiversity research in the Lake Hula area on an initiative basis. The algae species of the Hula Lake before it is drainage were studied and described by T. Rayss and E. Katchalsky [9], T. Rayss [10,11], and B. Komarovsky [12]. A. Ehrlich [13], and J.W. Sherman and R. Patrick [14] studied the diatom flora from the lake sediments. Unfortunately, in those two studies, the distinction between fossil and recent species was not always clear. In 1998, the list of algae species of Hula Lake was published [2]. The list summarized data of 1938–1958 and did not describe the algae flora from the Hula Nature Reserve. The studies of algae flora in the Hula Nature Reserve, funded by Israel Nature and Parks Authority, existed in 2007–2013 and included chlorophyll concentration measurements and analysis of algae and cyanobacteria species composition in samples collected from several sampling stations. As a result of monitoring, the main algae species of Hula Nature Reserve were determined. Unfortunately, the algae and cyanobacteria monitoring data remained "thing in itself", with a lack of ability to interpret data for understanding the ecological situation in the nature reserve state and change.

The European Water Framework Directive (WFD) requires an assessment of both the abundance and taxonomic composition of the aquatic communities [15]. The bioindication analysis is based on the hierarchical organization of the biotic community, which is described by the model of the trophic pyramid [16]. The distribution of the groups of organisms or species over the intervals of environmental factors is also of considerable importance. The level of adaptation of the species and the community as a whole determines the relationship between algal biodiversity and environmental conditions.

Most of the trophic state indices are based on the chemical and productivity data of the lake, such as chlorophyll and phosphorus initially defined by Carlson [17]. Methods for assessing the trophic state of the lakes are of particular interest to us [18,19]. Nevertheless, it is difficult or not adequate to calculate this type of indices when we have the species list only. Even the trophic state indices developed for the largest close placed Lake Kinneret include chemical variables, phytoplankton biomass, and chlorophyll-a concentration that are not defined yet in the Hula Lake [20]. The studies of ecosystem state for Lake Kinneret, the largest close placed lake, include biochemical variables, phytoplankton biomass, and chlorophyll concentration. For Lake Hula, these data are lacking. Therefore, only using

the bioindication method makes it possible to reconstruct and characterize the Lake Hula ecosystem state. For the Lake Hula ecosystem, this turned out to be the only way out in the analysis of long-term changes. Bioindication is based on the principle of congruence between community composition and the complexity of environmental factors. More of them, ecology of inhabitants can help to assess such factors as nutrition type correlated with ecotoxicology and trophic state of the water body. In the absence of environmental data and the available taxonomic composition of algae and cyanobacteria, it is possible to assess the change in water trophicity using the Nygaard index [21]. Therefore, we decided to implement bioindication methods for the assessment of long-term changes in the aquatic ecosystem of the protected Hula Lake.

We hypothesized that bioindication through algae and cyanobacteria species list with the relative abundance scores coupled with statistical methods could be used to assess the changes in biodiversity, water quality, and trophic state during the last hundred years in the Hula Nature Reserve.

The aims of the present study were:

- To compile the list of algae and cyanobacteria species from the past century references and our monitoring data and analyze it by bioindication and statistics;
- To compare the algae and cyanobacteria community of Lake Hula before drainage with it in the Hula Nature Reserve;
- To actualize the bioindication methods for monitoring the ecological situation in the aquatic ecosystem.

## 2. Materials and Methods

### 2.1. Description of Study Site

Lake Hula was formed about 20,000 years as a result of late Pleistocene volcanic activity. Basaltic hills intercept the Jordan River, form the "plug" restricted water drainage downstream into the Lake Kinneret, and separate the northern part of the Dead Sea Rift Valley that clearly defines Hula Valley (Figure 1).

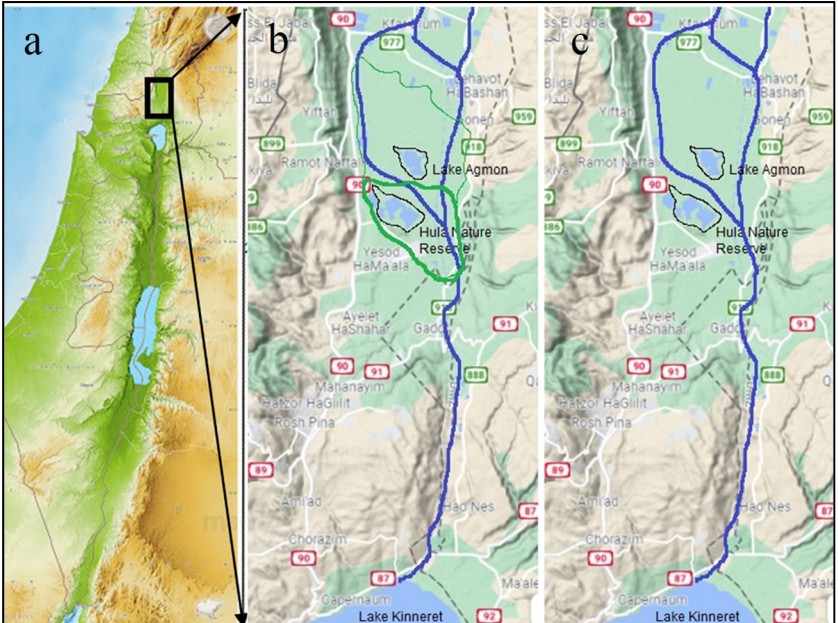

**Figure 1.** The Hula Lake historical changes during the last century. The Hula Nature Reserve on the Eastern Mediterranean (**a**); the Hula Valley before 1958 modified according to [22,23] (**b**), old Hula lake outlined by bold green line, Hula swamp area outlined by thin green line; the Hula Valley with Hula Nature Reserve, 1959–2021 modified according to [4,24,25] (**c**).

Appearing after the damming, Lake Hula became the base of the unique wetland ecosystem [6]. Prior to its drainage in the 1950s, Lake Hula was 5.3 km long and 4.4 km wide, extending over 12–14 km$^2$, depending on the water level. The lake depth fluctuated from about 1.5 m in summer to about 3 m in winter. The north part of the Lake Hula was extensive peat-based *Cyperus papyrus* swamps jointly covering an area of 45–75 km$^2$ (depending on seasonally fluctuating water levels) with several deeper open-water ponds [2,6]. Most of the water flowing into Lake Hula came from Mount Hermon via the Jordan River. Other streams, from the Golan Heights, the eastern Galilee Mountains, and from approximately 70 springs in the Hula Valley itself, also fed the lake and swamps. From the lake's outlet, the Jordan River flowed another 20 km before emptying into Lake Kinneret. The area is typically Mediterranean, with hot, dry summers and cool, rainy winters. However, the mountain-enclosed topography of the Hula Valley leads to more extreme seasonal, as well as daily, temperature fluctuations. Annual rainfall varies greatly between different parts of the valley and ranges from about 400 millimeters in the south to up to 800 millimeters in the north. More than 1500 millimeters of precipitation falls on the Hermon mountain range (mostly in the form of snow), feeding underground springs, including the sources of the Jordan River, and giving rise to much of the abundant water flowing through the valley. The wind regime is dominated by regional patterns in the winter, with occasional strong northeasterly windstorms. In summer, local warming and cooling patterns produce strong westerly to northerly winds in the afternoons. Due to its shallowness, the water temperature of Lake Hu1a exhibited large dial fluctuations, sometimes of more than 10 °C. In summer, typical afternoon westerly to northerly winds destroy the vertical temperature gradient, which developed during the calm morning hours. At night, convective mixing caused vertical turnover of the water layers. Monthly mean temperatures ranged from 12 to 27 °C, although much more extreme temperatures (4–38 °C) were also reported [2]. Not much information is available about the chemistry of the Lake. No brackish water sources fed into Lake Hula; therefore, its salinity was lower than that of Lake Kinneret, with chloride concentrations of 15–50 mg L$^{-1}$ and total dissolved solids 224–373 mg L$^{-1}$. The pH of the water ranged from 7.2 to 8.6. Of the inorganic nitrogen compounds, only ammonia was present in measurable concentrations (0.06 ppm), while nitrates were at undetectable levels. No data are available on P concentrations. The soil of the peat swamps contained 50–80% organic matter, mainly decomposed papyrus. Lake Hula and its adjacent swamps provided the habitat for a rich diversity of plants and animals and were a key feeding station for migratory birds. Unfortunately, anopheles (malaria) mosquito was a common part of the swamp fauna. In addition, according to the concept of the first half of the 20th century, the swamp occupied a significant area suitable for agriculture. Thus, in 1951 the works for drying the region, carried out by the Jewish National Fund, started, and in 1958, the Hula Lake and swamp were drained. Despite the complete transformation of the landscape, the drainage project included some conservation efforts. A 0.3 km$^2$ area known as the Hula Nature Reserve (HNR) was established in the southern end of the wetland area in order to maintain the existence of native flora and fauna. In addition to the HNR, the Einan Reservoir was constructed in 1984 to store and recycle effluent from fishponds and ultimately reduce nutrient outflow [26].

### 2.2. Sampling and Laboratory Studies

Samples for algae and cyanobacteria identification were collected monthly at stations 1, 4, 6, 8, and "Enan stream" (Figure 2). The 100 mL water samples for algae and cyanobacteria species analysis were fixed with 0.4 mL of Lugol solution and transferred for analysis at the Kinneret Limnological Laboratory (KLL).

At the Kinneret Limnological Laboratory, 10 mL of the Lugol-fixed samples were examined after 24 h sedimentation in sedimentation chambers under an inverted microscope, Axiovert 135M, Zeiss. For more details, fresh material was examined and photographed with an Olympus BX50 microscope equipped with a Nomarski DIC system with ×20, ×40, and ×100 DIC objectives (ECHO a BICO company, San Diego, CA, USA) and a PixeLINK

PL-662 digital camera (Pixelink (r)a Navitar Company. Gloucester, Canada). Algae were identified to genus or species level using the conventional taxonomic handbooks [27–33]. To avoid the use of synonyms, the identified species were checked in www.algaebase.org (accessed on 21 September 2021) [34]. All species were recorded and scored according to their relative abundance in the sample using the species frequencies scale (Table 1) [35,36].

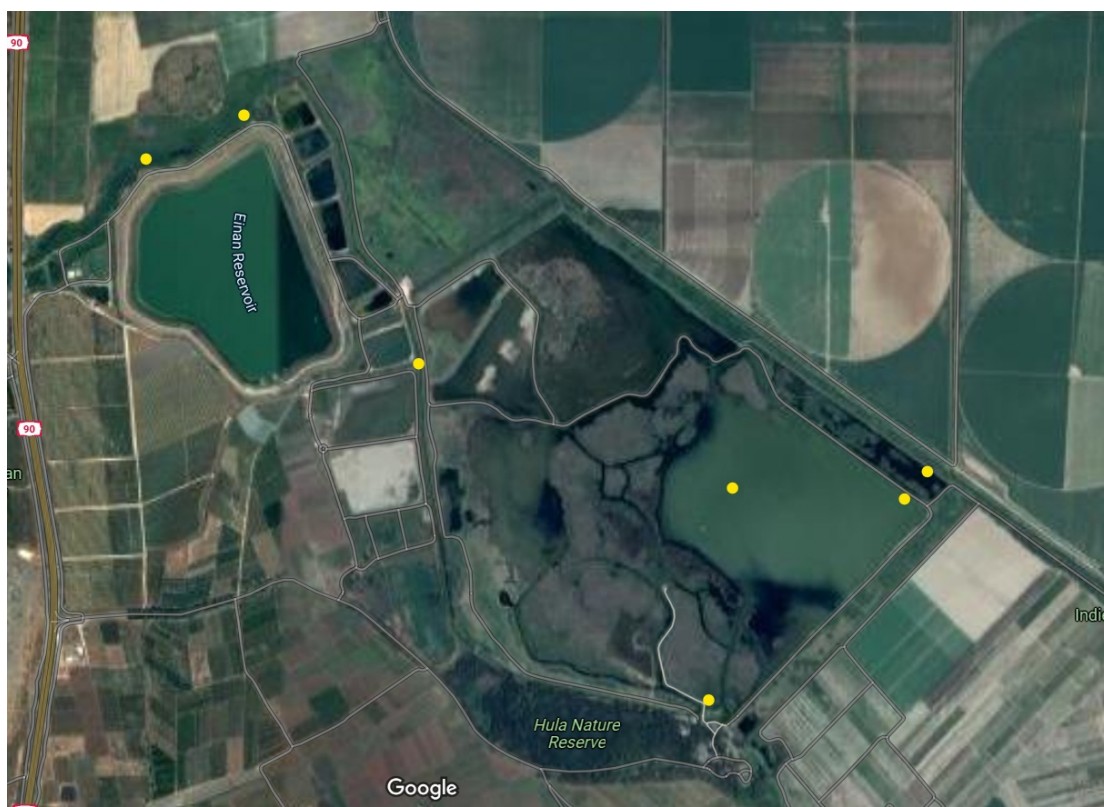

**Figure 2.** The Hula Lake/Hula Nature Reserve sampling sites in 1938–2013, yellow dots.

**Table 1.** Species frequencies scale according to the works of [35,36].

| Number of Observed Cells | Estimate | Score |
|---|---|---|
| 1–5 cells/chamber | Occasional | 1 |
| 10–15 cells/chamber | Rare | 2 |
| 25–30 cells/chamber | Common | 3 |
| 1 cell over a transect | Frequent | 4 |
| Several cells over a transect | Very frequent | 5 |
| A cell in every field of view | Abundant | 6 |

*2.3. Determination of Water Physicochemical Properties*

The major physicochemical variables of water quality (temperature, pH, electrical conductivity, total dissolved solids) were measured in parallel with algal sampling in 2011 at each sampling station on the spot by using HANNA HI-98194 multiparameter portable water quality meter.

*2.4. Historical Data Analysis*

We collected all available references containing the algae and cyanobacteria diversity data before the Lake Hula drainage. Because the sampling date was usually not clear, we used the paper publication year as the sampling date. The historical data comes from 1938 to 1958 was combined with our unpublished monitoring data comes from 2007 to 2013. The total species list was compiled and updated with modern taxonomy [34]. All

species list was divided into three periods: (1) Hula before draining actually based on the references of 1938–1958; (2) Hula after draining, Hula Nature Reserve existing but no any data 1959–2006; (3) Hula Nature Reserve (HNR) monitoring, 2007–2013.

### 2.5. Bioindication Analysis

The bioindication methods for a range of environment variables by the ecological preferences of the revealed algae and cyanobacteria species [37] and their abundance were used for analysis [38]. The bioindication properties of each revealed species came from a world database [39] compiled by us. The bioindication systems included the most effective indication of the organisms' such as substrate and nutrition type preferences, water salinity, alkalinity, organic pollution, and trophic state. The correlations between major indicative variables are given in the work of [38]. The saprobic index S was calculated according to V. Sládeček [40,41] to estimate the level of organic pollution. Index values S range from 0 (no polluted) to 4.5 (very polluted) for the aquatic environment. All data were ranked according to the CIS countries' classification system [42] to assess the water quality.

### 2.6. Statistical Analysis

The similarity of species richness was calculated in the BioDiversity Pro program. The correlation was calculated by network analysis in JASP Program [43].

### 2.7. Nygaard Species Indices for Trophic State Assessment

Nygaard [21] has developed trophic state-related indices (NY) based on large experimental observations of the diversity and species richness of the lake's phytoplankton. This became possible because both the diversity and abundance of species in taxonomic groups control the functioning of the ecosystem [44] and, thus, can reflect the trophic state of the lake. On the same theoretical basis, the Modified Nygaard Species Index (NS) and the sub-index of Quality Group species (QG) were constructed using the species richness of certain taxonomic groups according to the work of [21] and related to water quality [45]. The use of these indices is the last opportunity to assess the trophic status of a lake when there is no data on the chemistry of its waters.

Indices can be calculated on the basis of species richness of the lake phytoplankton in certain taxonomic groups. Nygaard Species Index [21] (NY) calculation based on the species richness of phytoplankton in higher taxonomic group and related to waterbody trophic states. It includes five sub-indices, which can be classified in relation to the lake trophic state as in Table 2:

Index 1 (NY-1) = Myxophyceae/Desmidiaceae
Index 2 (NY-2) = Chlorococcales/Desmidiaceae
Index 3 (NY-3) = Centrales/Pennales
Index 4 (NY-4) = Euglenineae/(Myxophyceae + Chlorococcales)
Index 5 (NY-5) = (Myxophyceae + Chlorococcales + Centrales + Euglenineae)/Desmidiaceae

**Table 2.** Nygaard Species Index (NY) table for classification of trophic the lake state.

| Trophic State | Index 1 | Index 2 | Index 3 | Index 4 | Index 5 |
|---|---|---|---|---|---|
| Oligotrophic-dystrophic phase | 0 | 0.0–0.3 | 0 | 0 | 0–0.3 |
| Oligotrophic-acidotrophic phase | 0 | 0–0.1 | 0 | 0 | 0–0.1 |
| Oligotrophic | 0–0.4 | 0–0.7 | 0 | 0–0.2 | 0.25–1.0 |
| Mesotrophic | 0.1–0.5 | 0.2–0.6 | 0–0.75 | 0.1–1.0 | 1.1–1.2 |
| Slightly eutrophic | 0.8–1.0 | 0.7–1.0 | 0.2–1.5 | 0–0.2 | 2.0–2.25 |
| Moderately eutrophic | 1.4–2.0 | 1.25–1.4 | 0.6–1.7 | 0 | 3.5–4.4 |
| Eutrophic | 1.2–3.0 | 2.1–3.5 | 1.25–3.0 | 0 | 4.3–8.75 |
| Eutrophic of mixotrophic phase | 0.9–2.7 | 2.2–2.5 | 0.2–0.5 | 0 | 3.3–5.3 |

Nygaard Species Index (NS) and Quality Group species sub-index (QG) were modified by the authors of [46] for assessment of the lake trophic state on the base of modern algae and cyanobacteria taxonomy (Table 3).

Modified Nygaard Species Index (NS):

NS = (cyanobacteria + chlorophytes + euglenophytes)/chrysophytes + desmids

Quality Group species sub-index QG:

QG = 5 × chrysophytes + 2 × desmids + dinoflagellates + cryptophytes + diatoms

**Table 3.** Modified Nygaard Species Index (NS) and Quality Group species sub-index (QG) classification for assessment of the lake trophic state (according to the work of [46]).

| NS | QG | Trophic State |
|----|----|---------------|
| 0–2 | >60 | Ultraoligotrophic |
| 2–4 | 46–60 | Oligotrophic |
| 4–6 | 31–45 | Mesotrophic |
| 6–8 | 16–30 | Eutrophic |
| >8 | <16 | Hypertrophic |

## 3. Results

### 3.1. Water Chemistry

Information on the chemistry of waters in Lake Hula is extremely scarce since it still does not have a system of constant monitoring. Table 4 presents the known data from U. Pollingher [2], R.F. Jones [47], and A. Nishri [48] works summarized the period of 1938–1958 before the draining of the lake, as well as our own data for 2011, referring to the monitoring period. Thus, Table 4 presents combined data that were collected both from literary sources in the historical period and collected by us sporadically, only in the summer period of 2011, which allows us to determine both changes in some environmental parameters and the overall amplitude of fluctuations.

**Table 4.** Chemical variables variation in the Hula Lake/Hula Nature Reserve in 1938–1958 and 2011.

| Parameter | 1938–1958 [2,47,48] | 2011 Our Data | 2011 Average Our Data |
|-----------|---------------------|---------------|------------------------|
| Water temperature, °C | 4–36 | 19.9–25.7 | 22.50 |
| TDS, mg L$^{-1}$ | 224–373 | 255–1675 | 734.75 |
| Sulfate, mg L$^{-1}$ | ≤373 | nd | nd |
| Nitrate-N, mg L$^{-1}$ | nd | 1.6–7.7 | 5.67 |
| Ammonia-N, mg L$^{-1}$ | 0.06 | nd | nd |
| Chloride, mg L$^{-1}$ | 15–50 | nd | nd |
| pH | 7.2–8.6 | 7.1–7.9 | 7.61 |
| Electrical conductivity, ms cm$^{-1}$ | nd | 0.36–2.26 | 1.01 |

Note: nd—not determined

It can be seen that the water temperature in 1938–1958 corresponds to the climatic norm, and the data for the spring-summer period 2011 are included in the amplitude. The pH of the water was also fairly stable for both periods. Sulfates, ammonium, and chlorides were measured only for the historical period and did not go beyond freshwaters. Of interest is the change in TDS upward from 370 mg L$^{-1}$ in 1938–1958 to 1675 mg L$^{-1}$ in 2007–2013. Moreover, according to our data, this indicator value changed from station to station. Similarly, a significant amplitude was found for the electric conductivity in the monitoring period, the values of which varied at different stations from 0.36 to 2.26 ms cm$^{-1}$ that already correspond to slightly brackish waters. The values of the concentration of nitrate nitrogen also fluctuated in the monitoring period from 1.6 to 7.7 mg L$^{-1}$ at different stations, which corresponds to significant organic pollution and Class 5 of water quality. Thus, the analysis of the values of chemical parameters shows that the ecosystem of Lake Hula between the historical and

monitoring periods underwent changes in water salinity, and organic pollution can be associated with the water flowing into the lake.

### 3.2. Species Richness

As a result of the study, 225 species and intraspecific taxa belonging to eight phyla were included in the compiled 1938–2013 taxonomic list of Hula's algae and cyanobacteria (Table S1). To check the completeness of the knowledge of the flora of algae and cyanobacteria in Lake Hula, a Willis curve was constructed (Figure 3). It can be seen that the line trend does not completely cover the distribution line; therefore, the flora is still far from being exhausted, but the general evened nature of the distribution makes it possible to analyze the taxonomic composition.

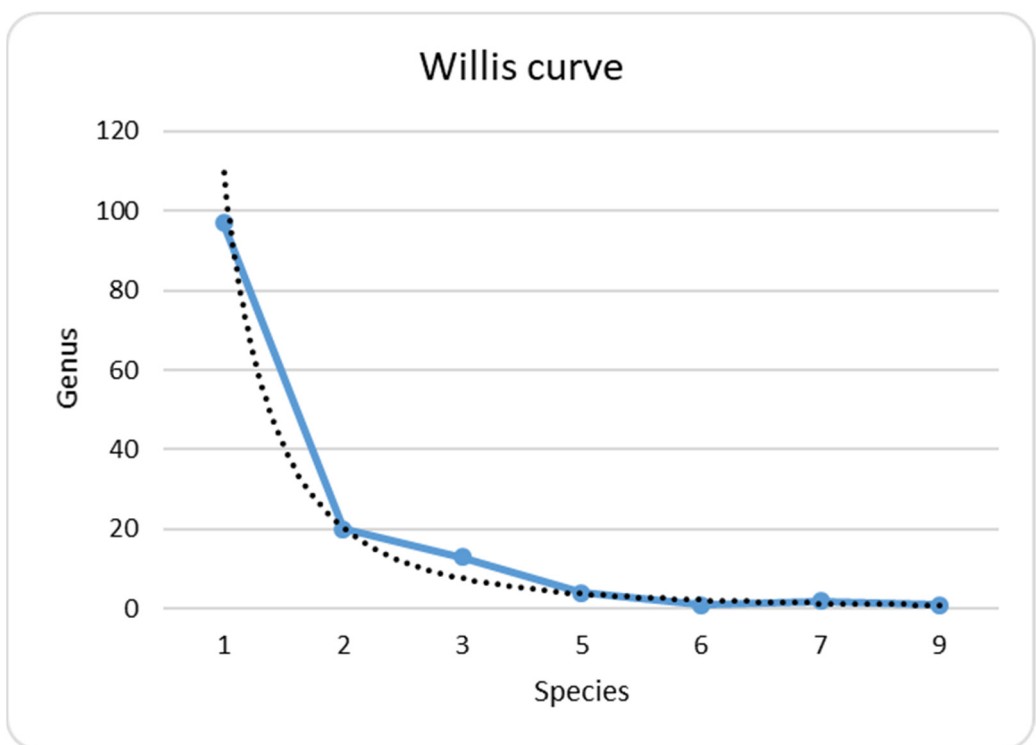

**Figure 3.** Willis curve for the Hula Lake/Hula Nature Reserve algae and cyanobacteria species revealed in 1938–2013.

Figure 4 shows the distribution of algae and cyanobacteria species richness over the study period. It can be seen that in the monitoring period, the number of species increased significantly in comparison with the first period (Table 5 and Table S1). At the same time, a similar distribution of the species composition according to phyla is observed in both periods with a noticeable predominance of green and diatoms; however, in the monitoring period, a significant number of cyanobacteria species also appear in aquatic communities. It can be seen that the most diverse and abundant was algae flora in 2009.

### 3.3. Bioindication

On the basis of species richness (Table S1), we revealed species-specific ecological preferences for each species during 1938–2013 and calculated its number in nine major indicator groups (Table 5). Most abundant were groups of Class 3 of water quality indicators, eurysaprobes, reflected medium oxygen enrichments, temperate temperature waters, preferred planktonic or plankto-benthic lifestyle, indifferent to salinity fresh and low alkaline waters, preferred photosynthetic type of nutrition but survive in the waters enriched by nutrients. Special attention can be given to the trophic state indicators distribution. It can

be seen in Table 5 that revealed algae and cyanobacteria species correspond to six from nine known trophic state groups. Eutrophic species strongly prevail.

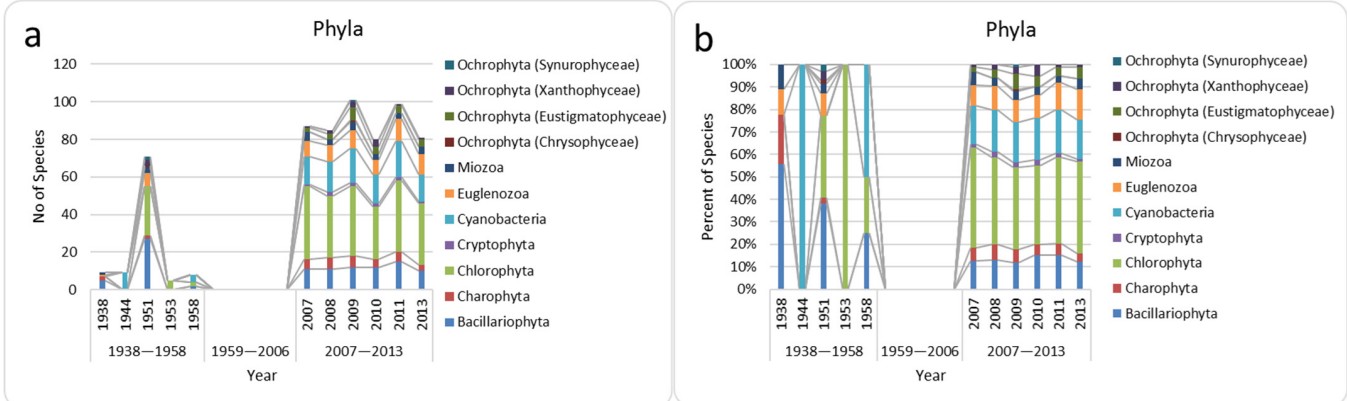

**Figure 4.** Distribution of the phytoplankton species number over taxonomic phyla in the Hula Lake/Hula Nature Reserve during the study period. (**a**) Species number, (**b**) percentage of species in phyla.

**Table 5.** Distribution of species richness in algae and cyanobacteria taxonomic phyla and indicator groups over 1938–2013 years in the Hula Lake and Hula Nature Reserve.

| Phylum | 1938 | 1944 | 1951 | 1953 | 1958 | 2007 | 2008 | 2009 | 2010 | 2011 | 2013 |
|---|---|---|---|---|---|---|---|---|---|---|---|
| | | Hula before drainage | | | | Hula Nature Reserve monitoring | | | | | |
| Bacillariophyta | 5 | 0 | 27 | 0 | 2 | 11 | 11 | 12 | 12 | 15 | 10 |
| Charophyta | 2 | 0 | 2 | 0 | 0 | 5 | 6 | 6 | 4 | 5 | 3 |
| Chlorophyta | 0 | 0 | 26 | 5 | 2 | 39 | 33 | 37 | 28 | 38 | 33 |
| Cryptophyta | 0 | 0 | 0 | 0 | 0 | 1 | 2 | 2 | 2 | 2 | 1 |
| Cyanobacteria | 0 | 9 | 0 | 0 | 4 | 15 | 16 | 18 | 15 | 19 | 14 |
| Euglenozoa | 1 | 0 | 7 | 0 | 0 | 8 | 9 | 10 | 8 | 12 | 11 |
| Miozoa | 1 | 0 | 3 | 0 | 0 | 5 | 3 | 4 | 3 | 3 | 4 |
| Ochrophyta (Chrysophyceae) | 0 | 0 | 1 | 0 | 0 | 0 | 0 | 1 | 0 | 0 | 0 |
| Ochrophyta (Eustigmatophyceae) | 0 | 0 | 0 | 0 | 0 | 2 | 3 | 7 | 4 | 4 | 4 |
| Ochrophyta (Xanthophyceae) | 0 | 0 | 3 | 0 | 0 | 1 | 2 | 3 | 4 | 1 | 1 |
| Ochrophyta (Synurophyceae) | 0 | 0 | 2 | 0 | 0 | 0 | 0 | 1 | 0 | 0 | 0 |
| **No of Species** | 9 | 9 | 71 | 5 | 8 | 87 | 85 | 101 | 80 | 99 | 81 |
| **Average Sum of Scores** | | | | | | 47.35 | 59.01 | 63.01 | 39 | 42.8 | 37.41 |
| **Index Saprobity S** | | | | | | 2.10 | 2.13 | 2.09 | 2.14 | 2.01 | 1.97 |
| **Class of Water Quality** | | | | | | | | | | | |
| Class 2 | 1 | 1 | 9 | 1 | 0 | 11 | 5 | 10 | 6 | 9 | 6 |
| Class 3 | 4 | 7 | 36 | 4 | 8 | 52 | 52 | 63 | 50 | 58 | 49 |
| Class 4 | 2 | 0 | 5 | 0 | 0 | 6 | 8 | 7 | 6 | 7 | 4 |
| Class 5 | 0 | 0 | 0 | 0 | 0 | 0 | 0 | 0 | 0 | 0 | 1 |
| **Watanabe** | | | | | | | | | | | |
| sx | 0 | 0 | 7 | 0 | 0 | 0 | 0 | 0 | 0 | 0 | 0 |
| es | 3 | 0 | 12 | 0 | 2 | 8 | 9 | 9 | 8 | 10 | 7 |
| sp | 1 | 0 | 1 | 0 | 0 | 1 | 1 | 1 | 1 | 1 | 1 |
| **Oxygen** | | | | | | | | | | | |
| aer | 0 | 0 | 0 | 0 | 0 | 2 | 1 | 1 | 1 | 1 | 2 |
| str | 0 | 0 | 0 | 0 | 0 | 0 | 0 | 0 | 0 | 0 | 0 |
| st-str | 7 | 0 | 48 | 5 | 5 | 45 | 45 | 49 | 41 | 50 | 45 |
| st | 2 | 0 | 19 | 0 | 2 | 10 | 11 | 12 | 9 | 12 | 10 |

**Table 5.** *Cont.*

| Phylum | 1938 | 1944 | 1951 | 1953 | 1958 | 2007 | 2008 | 2009 | 2010 | 2011 | 2013 |
|---|---|---|---|---|---|---|---|---|---|---|---|
| **Temperature** | | | | | | | | | | | |
| cool | 1 | 0 | 1 | 0 | 0 | 0 | 0 | 0 | 0 | 0 | 0 |
| temp | 3 | 0 | 17 | 0 | 1 | 7 | 7 | 7 | 7 | 10 | 6 |
| eterm | 1 | 0 | 3 | 0 | 2 | 5 | 6 | 6 | 4 | 6 | 5 |
| warm | 0 | 1 | 2 | 0 | 0 | 2 | 2 | 2 | 1 | 3 | 2 |
| **Habitat** | | | | | | | | | | | |
| B | 2 | 0 | 7 | 0 | 0 | 2 | 3 | 3 | 4 | 5 | 3 |
| P-B | 7 | 9 | 44 | 4 | 6 | 48 | 47 | 53 | 44 | 51 | 44 |
| P | 0 | 0 | 19 | 1 | 4 | 30 | 28 | 33 | 24 | 35 | 27 |
| **Salinity** | | | | | | | | | | | |
| hb | 0 | 0 | 3 | 0 | 0 | 2 | 1 | 1 | 2 | 2 | 2 |
| i | 6 | 3 | 45 | 5 | 6 | 41 | 38 | 41 | 34 | 41 | 34 |
| hl | 1 | 1 | 1 | 0 | 1 | 4 | 4 | 3 | 4 | 2 | 3 |
| mh | 1 | 0 | 3 | 0 | 1 | 1 | 5 | 4 | 4 | 4 | 4 |
| **pH** | | | | | | | | | | | |
| acf | 0 | 1 | 3 | 0 | 1 | 4 | 1 | 3 | 1 | 2 | 1 |
| ind | 3 | 2 | 21 | 2 | 3 | 21 | 23 | 25 | 24 | 22 | 18 |
| alf | 3 | 0 | 15 | 0 | 2 | 13 | 12 | 10 | 10 | 14 | 9 |
| alb | 0 | 1 | 0 | 0 | 0 | 2 | 1 | 2 | 3 | 3 | 2 |
| **Autotropy-Heterotrophy** | | | | | | | | | | | |
| ats | 0 | 0 | 7 | 0 | 0 | 1 | 0 | 1 | 1 | 1 | 1 |
| ate | 3 | 0 | 11 | 0 | 2 | 7 | 7 | 6 | 7 | 9 | 5 |
| hne | 1 | 0 | 2 | 0 | 0 | 3 | 4 | 4 | 4 | 3 | 3 |
| hce | 0 | 0 | 1 | 0 | 0 | 0 | 0 | 0 | 0 | 0 | 0 |
| **Trophic State** | | | | | | | | | | | |
| ot | 0 | 1 | 2 | 0 | 0 | 2 | 1 | 2 | 1 | 2 | 2 |
| om | 0 | 3 | 5 | 0 | 1 | 3 | 4 | 3 | 3 | 3 | 2 |
| m | 0 | 0 | 3 | 0 | 0 | 3 | 2 | 2 | 1 | 2 | 0 |
| me | 2 | 0 | 4 | 0 | 0 | 5 | 7 | 6 | 10 | 8 | 4 |
| e | 4 | 5 | 33 | 5 | 7 | 50 | 51 | 55 | 42 | 51 | 45 |
| o-e | 1 | 0 | 1 | 0 | 0 | 0 | 0 | 0 | 0 | 0 | 0 |

Note: **Habitat:** P— planktonic; P–B—plankto-benthic; B—benthic. **Temperature**: cool—cool water; temp—temperate temperature; eterm—eurythermic; warm—warm water. **Oxygenation and water moving (Oxygen):** st—standing water; str—streaming water; st-str—low streaming water; aer—aerophyles. **Halobity degree (Salinity):** i—oligohalobes-indifferent; hl—halophiles; hb—halophobes; mh—masohalobes. **Acidity (pH):** alf—alkaliphiles; ind—indifferents; acf—acidophiles; alb—alkalibiontes. **Organic pollution indicators according to Watanabe (Watanabe):** sx—saproxenes, es—eurysaprobes, sp—saprophiles. **Nitrogen uptake metabolism (Autotrophy-Heterotrophy):** ats—nitrogen-autotrophic taxa, tolerating very small concentrations of organically bound nitrogen; ate—nitrogen-autotrophic taxa, tolerating elevated concentrations of organically bound nitrogen; hne—facultatively nitrogen-heterotrophic taxa, needing periodically elevated concentrations of organically bound nitrogen; hce—facultatively nitrogen-heterotrophic taxa, needing elevated concentrations of organically bound nitrogen. **Trophic state:** ot—oligotraphentic; om—oligo-mesotraphentic; m—mesotraphentic; me—meso-eutraphentic; e—eutraphentic; o–e—oligo-eutraphentic. Classes of water quality toned by EU color code.

The distribution of indicator species over ecological groups in the studied period can help to reveal the environmental changes in Hula Lake over 1938–2013. Figures 5–9 show the bioindicators distribution over ecological groups in Hula before drainage and in Hula Nature Reserve. Figure 5 demonstrates increasing in planktonic species before drainage with stability after that. Temperature indicators were difficult to recognize in 1938–1958, but in 2007–2013 can be seen slightly increasing of eurythermic species that reflect warming waters.

Salinity indicators show water desalination in historical Hula, but indicators of saline waters contain about 20% and stay stable in the Hula Nature Reserve (Figure 6). pH indicators distribution was also difficult to interpret in 1938–1958, but in 2007–2013, the histogram showed continuously increasing alkalibiontes that reflect an increase in water pH that can be the result of groundwater inflow.

The distribution of oxygenation indicators (Figure 7) was rather stable with the prevalence of middle oxygenated waters before and after drainage. In Figure 6, the nutrition indicators distribution during 1938–1958 is difficult to interpret as a result of low indicator species number. At the same time, distribution in the monitoring period shows the prevalence of autotrophic species with up to 35% facultative heterotrophs.

Organic pollution diatom indicators, according to Watanabe (Figure 8), also contain not enough number in the historical period, but in a period of monitoring demonstrate stabile distribution where community contain only two groups, eurysaprobes and saptophiles in the same proportion and reflect middle polluted waters. Indicators of saprobity were a concern to dour classes with prevailing of Class 3. It can be seen (Figure 8) that the 1938–1958 self-purification process resulted in decreased indicators of Class 4 and increased species of Class 3. Distribution in 2007–2013 demonstrates the same tendency where indicators of Class 3 slightly increased, but the total proportion of indicators is very stable.

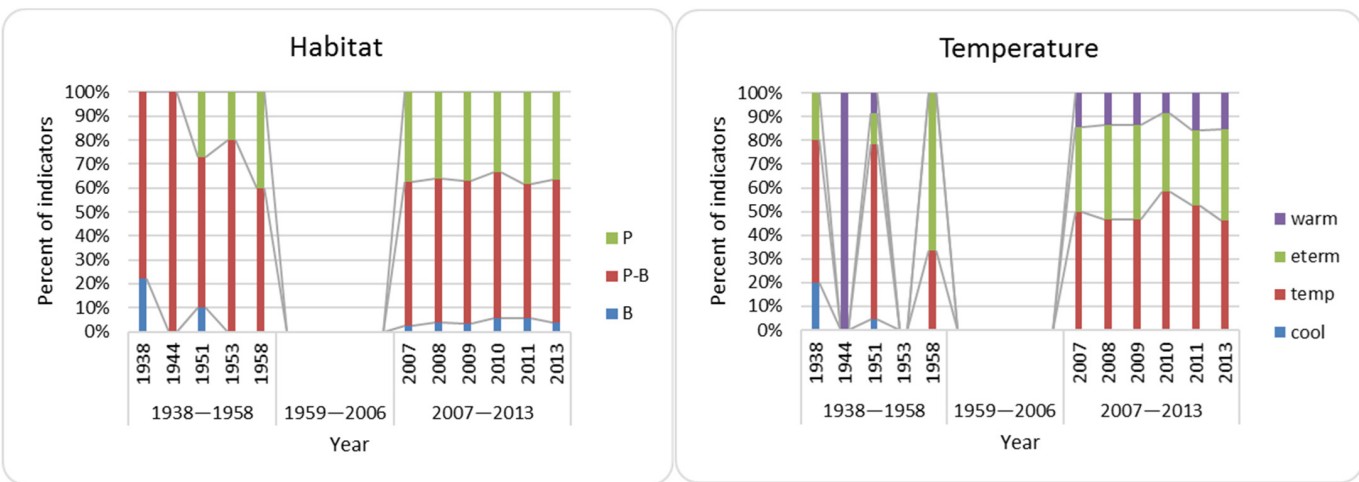

**Figure 5.** Distribution of the habitat preferences and water temperature indicators in the Hula Lake/Hula Nature Reserve algae flora during the study period. The ecological groups are arranged in order to indicate variable value ascending. Abbreviations: **Habitat**: P—planktonic; P–B—plankto-benthic; B—benthic. **Temperature**: cool—cool water; temp—temperate temperature; eterm—eurythermic; warm—warm water.

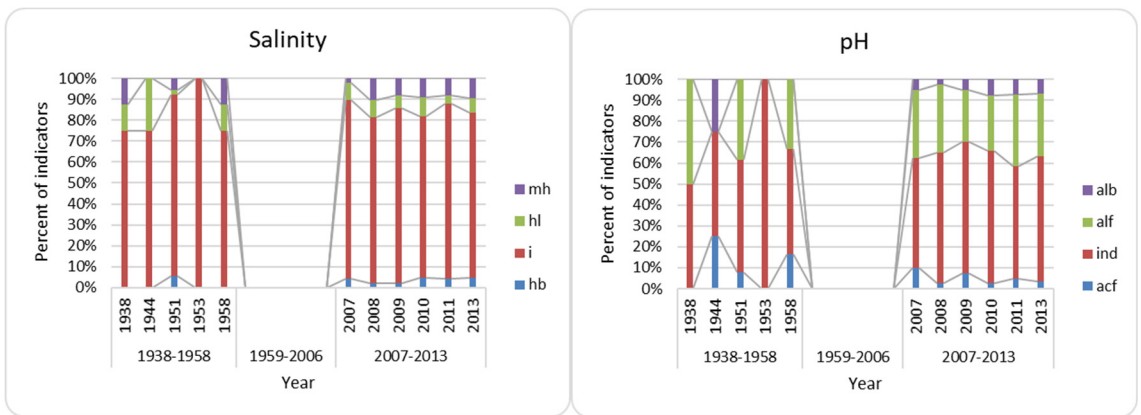

**Figure 6.** Distribution of the water salinity and pH indicators in the Hula Lake/Hula Nature Reserve during the study period. The ecological groups are arranged in order to indicate variable value ascending. Abbreviations: **Salinity:** i—oligohalobes-indifferent; hl—halophiles; hb—halophobes; mh —masohalobes. **pH:** alf—alkaliphiles; ind—indifferents; acf—acidophiles; alb—alkalibiontes.

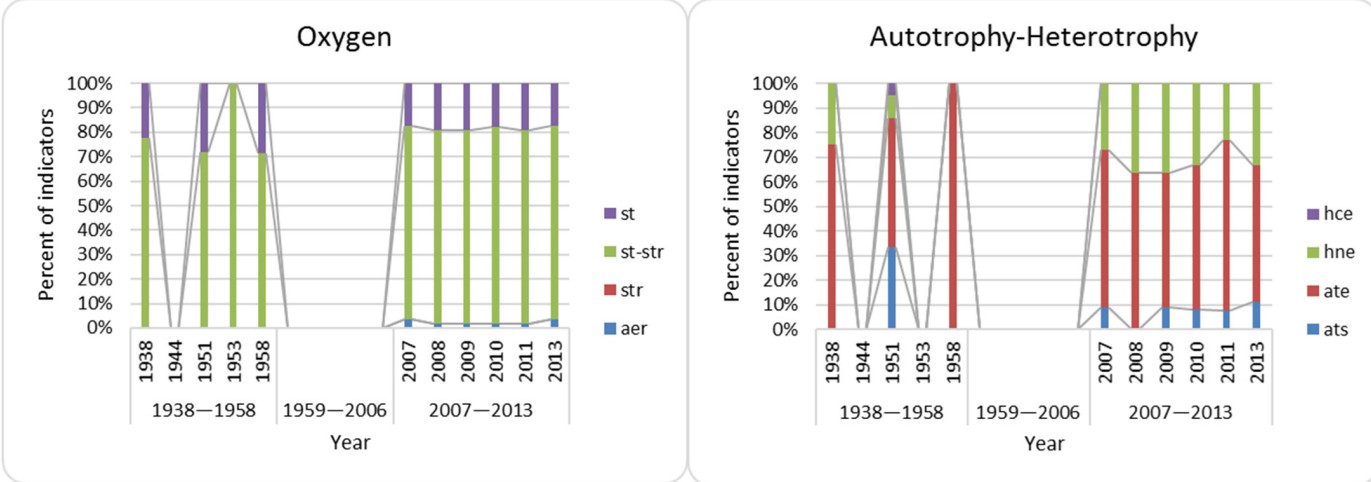

**Figure 7.** Distribution of the water oxygenation and the nutrition type indicators in the Hula Lake/Hula Nature Reserve during the study period. The ecological groups are arranged in order to indicate variable value ascending. Abbreviations: **Oxygenation and water moving (Oxygen):** st—standing water; str—streaming water; st-str—low streaming water; aer—aerophiles. **Nitrogen uptake metabolism (Autotrophy–Heterotrophy):** ats—nitrogen-autotrophic taxa, tolerating very small concentrations of organically bound nitrogen; ate—nitrogen-autotrophic taxa, tolerating elevated concentrations of organically bound nitrogen; hne—facultatively nitrogen-heterotrophic taxa, needing periodically elevated concentrations of organically bound nitrogen; hce—facultatively nitrogen-heterotrophic taxa, needing elevated concentrations of organically bound nitrogen.

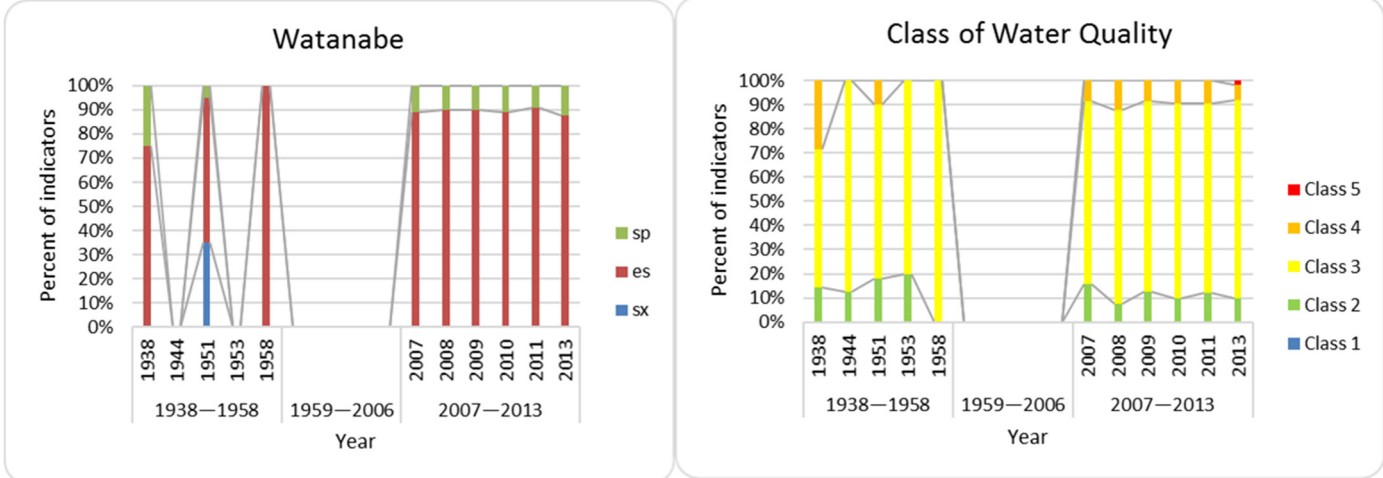

**Figure 8.** Distribution of the organic pollution indicators according to Watanabe and Class of Water Quality indicators in the Hula Lake/Hula Nature Reserve during the study period. The ecological groups and Class of Water Quality are arranged in order to indicate variable value ascending. Abbreviations: sx—saproxenes; es—eurysaprobes; sp—saprophiles. Colors of classes are in EU code.

The last distribution demonstrates the trophic state indicators dynamic (Figure 9). In 1938–1958, eutrophic indicators increased and continued their prevalence tater with increasing at the end of the studied period from 2010 to 2013 that reflect eutrophication process.

### 3.4. Statistical Analysis

We try to reveal some differences in species richness distribution of phytoplankton in Hula Lake during the study period. The tree of similarity in Figure 10 was constructed on the basis of species richness in each year (Table S1) and showed two major clusters, the first of which included species of the monitoring time with the addition of the 1951 year. The second cluster combined all other historical data. Therefore, the most important factor for

species similarity can be species richness. It means that important to continue to monitor the phytoplankton in Hula Lake to reveal the tendency in species richness succession.

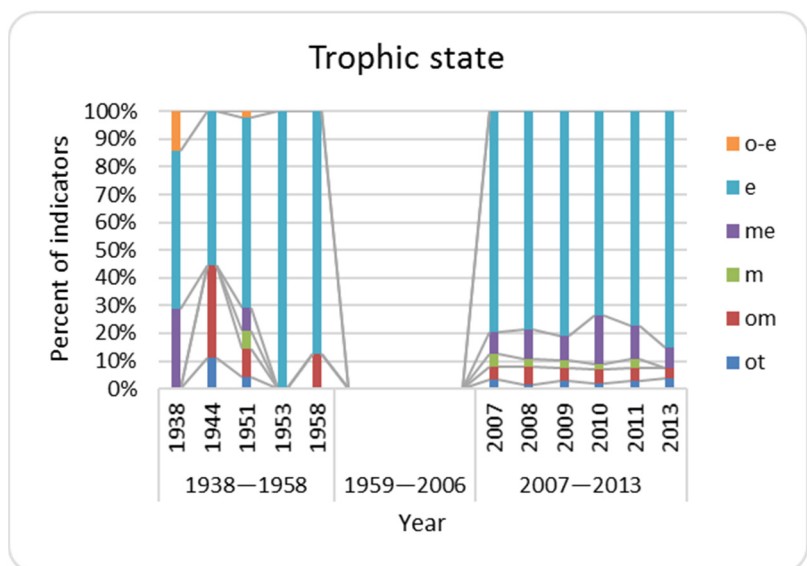

**Figure 9.** Distribution of the trophic state indicators in the Hula Lake/Hula Nature Reserve during the study period. The ecological groups are arranged in order to indicate variable value ascending. Abbreviations: ot—oligotraphentic; om—oligo-mesotraphentic; m—mesotraphentic; me—meso-eutraphentic; e—eutraphentic; o–e —oligo-eutraphentic.

Correlation for the JASP network (Figure 11) was calculated on the basis of Table 5, which included data of the distribution of species richness in phyla and environmental bioindicators groups over the years in studied periods. Can be seen two clusters, the first of which (cluster 1) combines variables from 1938 to 1958, while the second cluster includes all data from 2007 to 2013. Such a strict division by periods may be the result of the "alignment" of ecological and taxonomic indicators corresponding to different ecosystems separated in space and time.

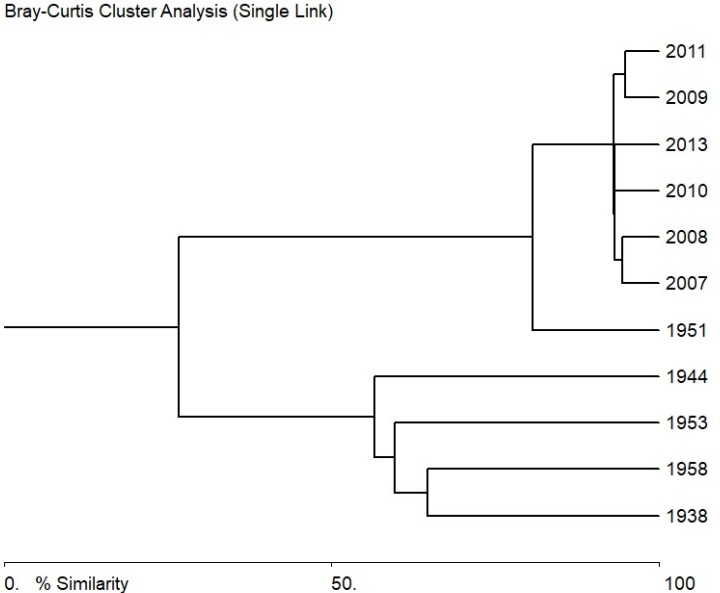

**Figure 10.** Tree of similarity by Bray–Curtis analysis of the Hula Lake/Hula Nature Reserve species in communities during 1938–2013.

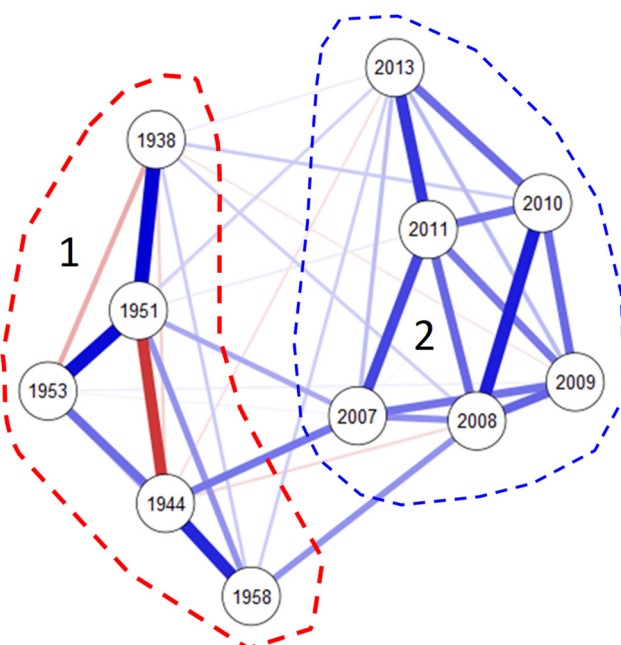

**Figure 11.** JASP network graph of correlation of species richness in algae and cyanobacteria Phyla and environmental bioindicator groups over the years in the Hula Lake/Hula Nature Reserve during 1938–2013. The line thickness is related to the value of the calculated correlation. Red lines reflect a negative correlation, and blue lines are positive correlations. Dashed lines outlined two different clusters of the Hula Lake ecosystem.

### 3.5. Index Saprobity S and Organic Pollution

Based on the known species-specific saprobity indices (Table S1) and data on the occurrence of algae and cyanobacteria species, which were determined only in the monitoring period, the saprobity indices were calculated for each of the communities. Then we calculated the average saprobity index for the year, which is presented in Table 5. The saprobity index S reflects the organic pollution of lake waters. The dynamics of the index S values are presented in Figure 12, along with the average species richness and the average sum of abundance scores for each year of monitoring. Figure 12 shows that index S fluctuated in the narrow range of Class 3 between 1.98 and 2.19. Nevertheless, the S values reflected the same Class of water quality; the trend line decreased and proposed organic pollution press during the monitoring period. The same decreasing tendency demonstrated species richness and algae and cyanobacteria abundance as the sum of scores. It can be the beginning of the negative succession of communities under the organic pollution press.

### 3.6. Trophic State Indices

The indices of the trophic status of the lake were calculated based on the distribution of the identified species composition over the main taxonomic groups. Table 6 shows the calculated values of the Nygaard Species Index and its modified variants made based on Table S1.

For the historical period, it was possible to calculate the NS index for 1951 only, when the algae and cyanobacteria community description was most complete and included all groups necessary for calculation. For 2007–2013, the NS index was calculated for each year. It turned out that trophicity of the Hula Lake according to NS index changed from oligotrophic before drainage to hypertrophic in the monitoring period. At the same time, the QG index turned out to be available for calculation, although species-poor communities during 1938–1958 excluding 1951 showed a hypertrophic state, which is clearly inadequate due to the calculation method. In 1951, the QG index assessed the lake as oligotrophic, and in 2007–2013 as eutrophic, or in 2009 as mesotrophic.

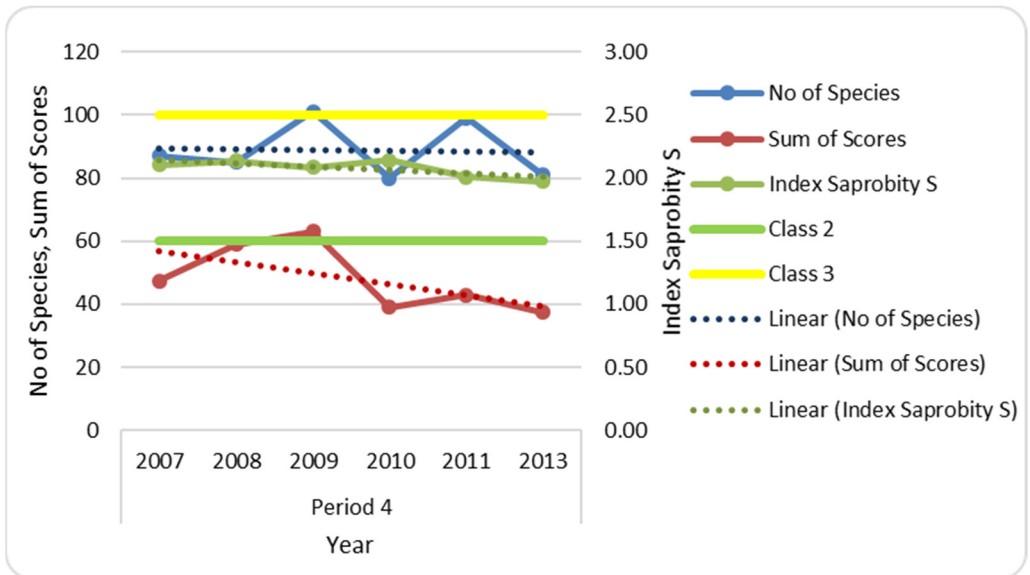

**Figure 12.** Distribution of the Hula Lake/Hula Nature Reserve algae and cyanobacteria species richness, abundance (as sum of scores), and calculated Index of Saprobity S for 2007–2013.

**Table 6.** Nygaard Species Index (NY), Modified Nygaard Species Index (NS), and Quality Group species sub-index (QG) calculated for the Hula Lake/Hula Nature Reserve algae and cyanobacteria communities in 1938–2013. Trophic state assessment category that can be used in monitoring are in bold.

| Index | Hula before Drainage | | | | | Hula Nature Reserve Monitoring | | | | | |
|---|---|---|---|---|---|---|---|---|---|---|---|
| | **1938** | **1944** | **1951** | **1953** | **1958** | **2007** | **2008** | **2009** | **2010** | **2011** | **2013** |
| **NS** | na | na | 6.6 | na | na | 12.4 | 11.6 | 9.3 | 12.8 | 17.3 | 29.0 |
| Trophic state | na | na | e | na | na | hy | hy | hy | hy | hy | hy |
| **QG** | 6 | 0 | 48 | 0 | 2 | 27 | 26 | 38 | 25 | 28 | 19 |
| Trophic state | **hy** | **hy** | **o** | **hy** | **hy** | **e** | **e** | **m** | **e** | **e** | **e** |
| **NY** | | | | | | | | | | | |
| index 1 | na | na | 0 | na | na | 3 | 3.2 | 3.6 | 3.75 | 4.75 | 7 |
| index 2 | na | na | 11 | na | na | 7.6 | 6.4 | 7 | 6.75 | 8.75 | 16 |
| index 3 | 0.67 | na | 0.24 | na | 1 | 0.38 | 0.57 | 0.5 | 0.33 | 0.36 | 0.43 |
| index 4 | 0.5 | 0 | 0.32 | 0 | 0.4 | 0.15 | 0.19 | 0.19 | 0.19 | 0.22 | 0.24 |
| index 5 | na | na | 17 | na | na | 12.8 | 12.2 | 13.4 | 13.25 | 17.5 | 30 |
| **Trophic state** | | | | | | | | | | | |
| index 1 | na | na | o | na | na | e | e | na | na | na | na |
| index 2 | na | na | na | o | na | na | na | na | na | na | na |
| **index 3** | **m** | **na** | **m** | **o** | **se** | **m** | **m** | **m** | **m** | **m** | **m** |
| **index 4** | **m** | **o** | **m** | **o** | **m** | **o** | **o** | **o** | **o** | **m** | **m** |
| index 5 | na | na | na | na | na | na | na | na | na | na | na |

The NY index for each of the five sub-indices was calculated for historical and monitoring data for individual sub-indices and years. In general, the change in the trophic status of the lake according to the NY index is assessed as a transition from mesotrophic or, in some years, oligotrophic before draining to oligotrophic at the beginning of monitoring time with a transition to mesotrophic at its end.

In general, it can be concluded that the QG index and the NY index, in the case of calculating sub-indices 3 and 4, can be used to assess the trophic status of Lake Hula, but only if algae and cyanobacteria communities are studied during the monitoring process, and not sporadically.

## 4. Discussion

The state of aquatic ecosystems is most often assessed by the chemical composition of water [37,49,50]. However, biological indicators can seriously complement chemical analysis. The bioindication method is integral and allows one to determine the dynamic of ecosystem development under the influence of the sum of factors, and not only to quantitatively determine a given physical or chemical parameter at the time of sampling [39,40,42,51–54]. In addition, the use of bioindicators can help in cases where chemical analysis data are not available [55], as in our case when a small set of chemical data was published for a historical period [2,47,48].

We applied the bioindication method to analyze historical data from Lake Hula and, despite the lack of synchronous chemical data, were able to follow up on the dynamic of the wetland ecosystem state.

Nevertheless, despite sporadically studying algae and cyanobacteria species composition in the Hula Lake and Nature Reserve in a period of about a hundred years (1938–2013), 225 species and intraspecific taxa belonging to eight phyla were revealed. Our calculations confirmed that the flora is still far from exhaustion and can be enriched with subsequent monitoring. This is a fairly large list of species for such a small territory and was sporadically studied compared to other wetlands in Europe, Israel, Ukraine, and Kazakhstan [50–54]. The same problem with the unevenness of research was successfully solved in the study of long-term changes in the lake using a list of algae and cyanobacteria species from publications of previous years [55] on the territory of Ukraine.

The face of the Hula Natural Reserve flora can be considered the predominance of green algae, followed by diatoms and cyanobacteria. Our analysis showed that this ratio remains constant in Hula Lake and HNR. However, the total species richness is incomparably higher in the HNR, most likely as a result of monitoring and detailed taxonomic analysis. It should also be noted that during regular studies in 2007–2013, the species richness and abundance changed synchronously, and the ratios in the taxonomical and ecological groups remained fairly constant, which may indicate the stability of the ecosystem and the success of the conservation regime.

We found ourselves in a difficult position in determining changes in the ecosystem of Lake Hula over such a long period since we did not have synchronous data on environmental parameters with data on algae and cyanobacteria communities. As is known, both data sets are needed to determine the state of an ecosystem [56], especially for the purpose of conservation and management of lakes [51,57]; moreover, it is chemical data that usually receive more attention in assessing ecosystem dynamics [58]. Indices showing changes in trophic status are mostly based on hydrochemical variables [59–61], which in our case is practically not available. We only have data on the species composition and even similarity of the year's species lists is important for revealing the differences between communities [62]. Despite the problem, we tried to assess the dynamics of the lake's ecosystem by establishing the environmental parameters by the communities of organisms living in it, using bioindication methods.

Although it was difficult to establish some indicator groups in 1938–1958 because of low species content, we revealed an increasing percentage of planktonic species in the historical period with stability in a period of monitoring that could be the result of the water body volume stabilization. Temperature indicators demonstrated a slight increase in eurythermic species that reflect warming waters due to significant water swallowing and a decrease in the area of the reservoir. In this regard, we confirmed that the Hula Nature Reserve is considered to the subtropical lakes more than to boreal [58,63]. In this case, it was very important to reveal some indicators that characterize the warm-water environment of the lake.

Salinity is an important variable in the semi-arid climate. Our analysis shows water desalination during 1938–1958, but in 2007–2013, it stayed stable. Water pH slightly increased, which can be the result of groundwater inflow, whereas the middle oxygenated waters indicators group prevailed during both periods and confirmed the stability of

the waterbody. We analyzed some variables that cannot be defined with the chemical analysis but are very important for the ecosystem characteristic. It means nutrition type of planktonic species and trophic state indicators. So, indicators in the Hula Lake were mostly autotrophic species with one-third of facultative heterotrophs. It means that the lake ecosystem periodically can be stressed by environmental factors but successfully avoid it. The analysis of a few aquatic ecosystems such as the Lower Jordan [64], Qishon [65], and Hadera [66] rivers, and especially the Upper Jordan River catchment basin [58,67,68] shows that ecosystem stress by nutrients inflow, toxic substances, salinity or temperature can be revealed more effective by bioindicators than by chemical methods.

The trophic state indicators distribution shows increasing in eutrophic species number in time and reflects the eutrophication process. In this relation, we analyzed the water quality indicators distribution over the studied periods and revealed that species of Class 3 slightly increased, but the total proportion of indicators of water quality is very stable in the old Hula as well as in the HNR.

Comparison of species richness of algae and cyanobacteria in the Hula Lake during study years with statistical methods reveal two different clusters, the first of which combined taxa of 2007–2013 with the addition of the 1951 year and second with other historical data. This difference in distribution helps us to allow that the most important factor for species similarity can be the number of species in the community. It confirms that important to continue the algae and cyanobacteria flora study in Hula Lake for revealing the tendency in succession. Statistical correlation of the phyla species richness and number of environmental indicators, on the contrary, revealed to be strong following to drainage periods that can be the result of the "alignment" of ecological and taxonomic indicators corresponding to different ecosystems separated in space and time.

The saprobity index S calculated by us for the monitoring years reflects the organic pollution of lake waters fluctuated in the narrow range of Class 3 with decreasing in time of 2007–2013. It can be evidence of the self-purification process in the Hula Nature Reserve ecosystem on the one hand, but the similar decreasing tendency of species richness and abundance (as the sum of scores) can reflect the beginning of the negative succession of the aquatic community as a result of organic pollution press.

In this case, it is important to reveal the trophic state tendency in the lake ecosystem. However, we have the same problem: we have the species list only. For calculation of trophic state indices (many of which are based on a combination of chemical data and aquatic community productivity), we choose the Nygaard index [21] and its modern derivates [46,69], where the taxonomic composition is enough for the trophic state assessment. The calculated indices of the trophic state make it possible to classify Lake Hula as one that changed from oligotrophic in the historical period to eutrophic or even hypertrophic in the 2000s. The QG and NY indices [69] can be recommended for assessing the trophic status of wetlands and aquatic ecosystems, but only if species composition had been studied during monitoring and not sporadically.

A comparison of the floras and bioindicators of the two Ramsar wetlands in Israel, the Hula Nature Reserve in the rift valley below sea level, and the Afek Nature Reserve [70] on the Mediterranean coast may also provide a direction for future research. The mechanism and criteria have been developed for the well-studied regions of the Mediterranean basin [71,72] in the context of climate aridization and anthropogenic transformation. Further areas of work may be the expansion of the use of bioindication in the monitoring of small water bodies in Israel for a better understanding of ecosystems' state and identification of potential objects of protection.

## 5. Conclusions

1. For the first time, for long-term bioindication analysis used compiled data from references previous decades published and modern monitoring results;
2. Despite the sporadical character of algae and cyanobacteria studies in the Hula Lake/Hula Nature Reserve, 225 species and intraspecific taxa belonging to eight

phyla were revealed in 1938–2013. This species list is fairly large for such a small territory but is still far from exhaustion and can be enriched with subsequent works;

3. Comparison of Lake Hula and Hula Nature Reserve algae and cyanobacteria flora reveals many similarities; however, the bioindication noticed salinity and organic pollution increase in a modern time. The increase seems to be caused by water flowing into the Hula Nature Reserve water body;

4. The use of bioindication methods and our own created database of aquatic flora species ecological preferences made it possible to mark the environmental dynamics of water bodies only based on the species composition of algae and cyanobacteria. Our work shows the special relevance of bioindication for monitoring aquatic ecosystems in situations of impossibility to obtain hydrological data and count the exact species abundance and biomass.

**Supplementary Materials:** The following are available online at https://www.mdpi.com/article/10.3390/d13110583/s1. Table S1. Diversity and ecology of algae and cyanobacteria in the Hula Lake/Hula Nature Reserve over 1938–2013.

**Author Contributions:** Conceptualization, S.B. and A.A.; methodology, S.B. and A.A.; software, S.B.; validation, S.B. and A.A.; formal analysis, S.B. and A.A.; investigation, S.B. and A.A.; resources, S.B. and A.A.; data curation, A.A.; writing—original draft preparation, S.B. and A.A.; writing—review and editing, S.B. and A.A.; visualization, S.B.; supervision, S.B.; project administration, S.B. All authors have read and agreed to the published version of the manuscript.

**Funding:** This research received no external funding.

**Institutional Review Board Statement:** Not applicable.

**Informed Consent Statement:** Not applicable.

**Data Availability Statement:** Data available in Table S1.

**Acknowledgments:** We are very thankful to Yifat Artzi for sampling, who did all fieldwork during the Hula Nature Reserve monitoring program and information help while writing the article. Israel Nature and Parks Authority funded the Hula Nature Reserve monitoring. This work was partly supported by the Israeli Ministry of Aliyah and Integration. We are also thankful to Elena Cherniavsky for help in the database compilation.

**Conflicts of Interest:** The authors declare no conflict of interest.

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
