# Peer review of "Algae and Cyanobacteria Diversity and Bioindication of Long-Term Changes in the Hula Nature Reserve, Israel"

_diversity, doi:10.3390/d13110583_

Round 1
Reviewer 1 Report
This is an extraordinarily detailed and careful ecological analysis of a significant ecosystem based on an ecological group, the phytoplankton, with high indicative value. Exemplary work!
Please check throughout the text for the use of the two terms Hula Lake and Hula Nature Reserve. It is not clear whether Hula Lake is used consistently when referring to the "old" lake in its broad extent, and/or the "new" small lake including wetlands or exclusive wetlands. For example, when you write in the abstract "The phytoplankton flora of Lake Hula and Hula Nature Reserve was found to be similar" it is not clear whether you mean by Hula Nature Reserve the totality of the 11 sampling sites. These are so diverse habitats, from springs, tributaries, the actual lake, they can't all be "similar" to the old lake in the last century. Therefore, I find the conclusion about the multiple similarity between the two study periods very ambitious. Overall, it would be interesting to know more about the diversity of the 11 sampled habitats.
line 44 add „forced by…“
line 187 Fig. 2 Please explain the different sampling sites
line 286: something is missing after "first" (period?)
Species list: Please clarify the identification of Arthrospira platensis. This taxon is important for indicating saline conditions. Is it Arthrospira fusiformis or Limnospira Nowicka-Krawczyk et al. 2019 (DOI:10.1038/s41598-018-36831-0)
Author Response
Responses to Reviewer 1
Dear Reviewer,
Thank you for your comments. All of them are inserted in the form of corrections in the text of the article and are marked in red.
With best regards,
Prof Sophia Barinova,
Corresponding author
This is an extraordinarily detailed and careful ecological analysis of a significant ecosystem based on an ecological group, the phytoplankton, with high indicative value. Exemplary work!
Please check throughout the text for the use of the two terms Hula Lake and Hula Nature Reserve.
Response: We used name “Hula Lake” or old Hula for the lake existed before drying in 1958. The wetland part of the Hula Lake left after the draying officially named “Hula Nature Reserve” so we used this name for the data obtained after 1958.
It is not clear whether Hula Lake is used consistently when referring to the "old" lake in its broad extent, and/or the "new" small lake including wetlands or exclusive wetlands. For example, when you write in the abstract "The phytoplankton flora of Lake Hula and Hula Nature Reserve was found to be similar" it is not clear whether you mean by Hula Nature Reserve the totality of the 11 sampling sites. These are so diverse habitats, from springs, tributaries, the actual lake, they can't all be "similar" to the old lake in the last century. Therefore, I find the conclusion about the multiple similarity between the two study periods very ambitious. Overall, it would be interesting to know more about the diversity of the 11 sampled habitats.
Response: Since in the part of the work devoted to the old Hula, we summarized the published data collected by the authors of the cited works in different parts of the lake, often without specifying a specific place, then, accordingly, in the part of the work concerning the Hula Nature Reserve, we considered it more correct to also present the summarized information. The station numbers in Figure 2 are related to the marking of the land plot and do not reflect the number of stations at which samples were taken, the total number of which is 8. Figure 2 is replaced by a similar one without station numbers, which are not related to this work, but only to the internal documents of the reserve Hula. We strongly agree that it is necessary to compare what is happening at the sampling stations, but this is part of the plan for further work with other analytical approaches and tools. Since the Hula reserve is part of a pre-existing lake, located in the same landscape and climatic zone on the same specific peat soils, we believe that the similarity, this is normal, corresponds to the status of a protected area and confirms the effectiveness of the conservation regime, while a sharp difference historically, it would be evidence of catastrophic anthropogenic changes. Precisely because we used all the available information about the algae of Lake Hula in all periods, we decided to change the title of the work to: Algae and cyanobacteria diversity and bioindication of long-term changes in the Hula Nature Reserve, Israel.
line 44 add „forced by…“
Response: corrected as: The Jewish National Fund carried out the draining project, yet its objectives were never achieved. Contrary, the draining led to uncontrollable underground fires that resulted in the formation of dangerous caverns within the peat layer and dust storms in the valley.
line 187 Fig. 2 Please explain the different sampling sites
Response: Fig 2 modified.
line 286: something is missing after "first" (period?)
Response: period
Species list: Please clarify the identification of Arthrospira platensis. This taxon is important for indicating saline conditions. Is it Arthrospira fusiformis or Limnospira Nowicka-Krawczyk et al. 2019 (DOI:10.1038/s41598-018-36831-0)
Response: Based on morphology and ecology we think that Arthrospira platensis is the right identification. The species was found in the summer period of different years sporadically, in low abundance. Despite that the species is defined as mesohalob, we think that it could be also found in freshwater habitats with high evaporation.

Reviewer 2 Report
In this manuscript, Barinova and Alster performed a present and retrospective characterization of the phytoplankton of the Hula Nature Reserve along with the study of several derived indices of trophism. To do so, they used a combination of past data and recent observations and performed a classical analysis of the microbiome of the samples based on microscope combined with different statistical analysis. Despite the study can be of interest, I have several remarks that should be address prior to publication in the journal. Moreover, despite I am not a native speaker, I find the English has room for improvement.
General comments:
The database used is not clear and should be better explained. Do they reinterpret the samples from 1938-1958 or the data they provide are based on past analyses? In addition, the year 1951 is clearly an outlier within this series. Was this sample collected at the same moment of the year? Fixed-conserved better? This represent a problem when trying to perform robust statistical analysis and correlations. The authors should discuss this (only mentioned in line 490 and not in depth).
The lack of molecular data is annoying. The characterization of phytoplankton based only on microscopic observations can be useful for a routine-control of the lake, but nowadays scientific production must be based on some molecular information.
Please see the pdf for a more line-by-line review.

Author Response
Responses to Reviewer 2
Dear Reviewer 2,
Thank you for your comments here and in pdf file. All of them are inserted in the form of corrections in the text of the article and are marked in red.
With best regards,
Prof Sophia Barinova,
Corresponding author
In this manuscript, Barinova and Alster performed a present and retrospective characterization of the phytoplankton of the Hula Nature Reserve along with the study of several derived indices of trophism. To do so, they used a combination of past data and recent observations and performed a classical analysis of the microbiome of the samples based on microscope combined with different statistical analysis. Despite the study can be of interest, I have several remarks that should be address prior to publication in the journal. Moreover, despite I am not a native speaker, I find the English has room for improvement.
Response: English editor checked paper text.
General comments:
The database used is not clear and should be better explained. Do they reinterpret the samples from 1938-1958 or the data they provide are based on past analyses? In addition, the year 1951 is clearly an outlier within this series. Was this sample collected at the same moment of the year? Fixed-conserved better? This represent a problem when trying to perform robust statistical analysis and correlations. The authors should discuss this (only mentioned in line 490 and not in depth).
Response: We analyzed the data we collected during 2007-2013, and also the historical data from published works in 1938-1958. All results were combined into a general list of species with a mention of the presence of each species by year. That is, the selected data set includes accessible and homogeneous information. Thus, we decided to change the title of the work to: Algae and cyanobacteria diversity and bioindication of long-term changes in the Hula Nature Reserve, Israel.
The lack of molecular data is annoying. The characterization of phytoplankton based only on microscopic observations can be useful for a routine-control of the lake, but nowadays scientific production must be based on some molecular information.
Response: Since we analyzed not only the data we collected, but also the historical data, there was no question of the molecular analysis of historical samples. That is, the data set we selected included, first of all, accessible and homogeneous information. Thus, we decided to change the title of the work to: Algae and cyanobacteria diversity and bioindication of long-term changes in the Hula Nature Reserve, Israel.
Please see the pdf for a more line-by-line review.
Response: All mentioned in pdf comments has been taken in account and corrected in main text of paper.

Round 2
Reviewer 2 Report
The authors have addressed all my concerns